# Stress Estimation Using Biometric and Activity Indicators to Improve QoL of the Elderly [note 1]

**DOI:** 10.3390/s23010535

**Published:** 2023-01-03

**Authors:** Kanta Matsumoto, Tomokazu Matsui, Hirohiko Suwa, Keiichi Yasumoto

**Affiliations:** 1Nara Institute of Science and Technology, 8916-5 Takayama-cho, Ikoma-shi 630-0192, Japan; 2RIKEN Center for Advanced Intelligence Project AIP, Tokyo 103-0027, Japan

**Keywords:** stress level estimation, daily living activity, biometric data, wearable sensors

## Abstract

It is essential to estimate the stress state of the elderly to improve their QoL. Stress states change every day and hour, depending on the activities performed and the duration/intensity. However, most existing studies estimate stress states using only biometric information or specific activities (e.g., sleep duration, exercise duration/amount, etc.) as explanatory variables and do not consider all daily living activities. It is necessary to link various daily living activities and biometric information in order to estimate the stress state more accurately. Specifically, we construct a stress estimation model using machine learning with the answers to a stress status questionnaire obtained every morning and evening as the ground truth and the biometric data during each of the performed activities and the new proposed indicator including biological and activity perspectives as the features. We used the following methods: Baseline Method 1, in which the RRI variance and Lorenz plot area for 4 h after waking and 24 h before the questionnaire were used as features; Baseline Method 2, in which sleep time was added as a feature to Baseline Method 1; the proposed method, in which the Lorenz plot area per activity and total time per activity were added. We compared the results with the proposed method, which added the new indicators as the features. The results of the evaluation experiments using the one-month data collected from five elderly households showed that the proposed method had an average estimation accuracy of 59%, 7% better than Baseline Method 1 (52%) and 4% better than Baseline Method 2 (55%).

## 1. Introduction

The proportion of the elderly in the population is increasing rapidly worldwide. This population transition to an “aging society” brings significant economic and social risks. Effective measures to promote the care of the elderly and the extension of healthy life expectancy are urgently needed. To promote the welfare of the elderly, it is necessary not only to support them in medical institutions, but also to help them live independently in their own homes. Therefore, it is essential to build an environment in which elderly people can understand their daily activities and manage their health conditions by themselves. There have been many studies (e.g., ref. [1]) on activity sensing/recognition technologies in the home, but studies on health condition monitoring to build an environment for self-health management and improve the lifestyle habits of the elderly still remain in the early phases.

In order to provide monitoring services and improve the lifestyle of the elderly, it is necessary to develop indicators of health status and investigate the factors that cause these indicators to change.

In past studies on health-related quality of life (HRQoL), many publications analyzed the relationship between health status and quality of life in various situations, including old and young, men and women, and with and without illness [2,3,4,5,6]. In the context of global aging, the quality of life of the elderly has been much studied in the context of extending healthy life expectancy [6,7,8].

The World Health Organization (WHO) has developed a questionnaire called WHOQOL-100 [9] as a representative indicator of health status. However, the method of measurement using such questionnaire indicators places a heavy burden on respondents due to the large number of questions.

For this reason, various stress estimation and prediction techniques have been studied. Amenomori et al. [10] conducted a study to measure HRQoL with less burden by using devices such as smartphones and smartwatches. They aimed to improve HRQoL, which is strongly related to physical and mental stress states, by continuously measuring HRQoL and detecting signs of stress early to prevent it.

Stress estimation has also been studied using devices rather than questionnaires to reduce the burden of estimation. Stress estimation using devices has been applied to many situations such as daily life [11,12,13]. Natasha et al. [12] proposed a method to quantitatively measure and estimate health, stress, and happiness on the following day using smart devices. Gjoreski et al. proposed an automatic detection of students’ stress using smartphones [13]. This approach aims to develop a machine learning model to detect students’ stress levels. They used data from multiple smartphones, including accelerometers, audio recorders, GPS, WiFi, call records, and light sensors.

While these studies used smartphones, recorded precise daily living activities, and measured screen time, they did not take into account the characteristics of the elderly, who are not accustomed to digital devices and tend to be strict about privacy, making the system highly burdensome for the elderly. This study created a low-burden stress estimation method because it used data on in-home daily living activities that can be automatically obtained by smart homes and biometric data that can be automatically collected by wearing a smartwatch. Aiming to improve stress estimation accuracy, in this paper, we propose a method to estimate stress levels by linking daily living activities’ data and biometric data. Specifically, we constructed a machine learning model with the results of a stress status questionnaire obtained every morning and evening as the objective variable and the features calculated from biometric data and daily living activity data as the explanatory variables. As the biometric data, we used the Lorenz plot area calculated from the heart rate data collected by a smartwatch (Fitbit), as it can visualize the activity level of the parasympathetic nervous system and has been shown to be useful for stress estimation [14]. Existing studies have used the heart rate, smartphone acceleration, and physical activity as explanatory variables in stress assessment models. We considered that there is a relationship between in-home daily living activities and stress and used the heart rate for each in-home daily living activity as an explanatory variable. The novelty of this study is that we used these explanatory variables to create a new indicator.

The study also proposes a new indicator that includes biological and activity perspectives as features. The reasons for using activity and biometric data for this are that other studies have examined the relationship between physical ability and QoL and physical activity levels and many studies have been conducted using biometric information, which can be acquired by wearable devices, to detect stress [15,16,17,18,19]. The physical approach is examining passive QoL and health measures from a machine learning perspective using “physical activity data” such as smartphone acceleration and tilt [20,21]. There is also a positive correlation between physical activity and QoL [15]. It is useful to predict QoL from physical capacity and physical activity and daily living activity data. The biological approach is based on a physiological analysis using signals collected from ECG [22], skin electrical activity [23], RRI [24], and myoelectric sensors [25].

To evaluate the effectiveness of the proposed method, we applied the method to the dataset [1] consisting of daily living data, biometric data, and stress status questionnaires (physical stress in the morning and evening) collected from five households of elderly people over 60 years old for 1 month. In our previous study [26], the stress estimation model was trained with the data of all of the five households. However, individual differences in the data were rather large, and the estimation accuracy remained low. Therefore, in this paper, we created a new indicator (called *mixed indicator*) by combining normalized biological and activity indicators to solve this problem.

For the evaluation, we compared the proposed method with two baseline methods: the Baseline Method 1, which uses 4 h after wakeup and 24 h before the questionnaire of the RRI variance and Lorenz plot area (shown to be effective in [14,27]) as features; Baseline Method 2 adds sleep duration (shown to be effective in [11]) as a feature to Baseline Method 1; the proposed method which adds the Lorenz plot area for each activity, the total time for each activity, and the mixed indicator for each activity as features to Baseline Method 2. As a result, the proposed method achieved an estimation accuracy of 59% on average, which is 7% better than Baseline Method 1 (52%) and 4% better than Baseline Method 2 (55%).

The main contributions of this paper are summarized as follows:A new indicator, including biological and activity perspectives, was developed and used as a feature to estimate the health status of the elderly.The estimation performance was improved by 4–7% by introducing the developed indicator and some analytical methods (e.g., bagging, upsampler, downsampler, and SHAP) compared to baseline methods and our previous study [26].

## 2. Related Research

This section surveys existing studies on QoL estimation, stress estimation, and health management using intelligent home technologies.

### 2.1. QoL Estimation

Quality of life (QoL) measures the satisfaction with and the quality of daily life. QoL [28] research originally started as a concept to discuss the quality of life after treatment in the medical field, but it is now used not only in the medical field, but also as a concept related to the quality of life in general, such as work–life balance and happiness. In particular, QoL, which is directly related to human health, is called health related-quality of life (HRQoL) and is evaluated by categorizing it into various domains such as physical, psychological, social interaction, economic and occupational, and religious and spiritual states. The World Health Organization (WHO) has developed various indicators to quantitatively assess HRQoL, such as WHOQOL [9] and the Short Form [29]. These indicators are assessed using paper questionnaires. However, WHOQOL-100 [9,30] requires 100 items in 6 domains, while SF-36 [29] requires 36 items in 8 domains. The labor to answer these questions makes it difficult to assess the quality of life on a daily basis. Prasad et al. [15] assessed the physical functioning of the elderly living in India to determine its association with the physical activity level and quality of life and found a positive correlation between physical functioning and quality of life. There was also a positive correlation between physical activity level and quality of life. Zapata-Lamana et al. [17] aimed to assess whether factors that foster the physical activity level and quality of life could be predictors of life satisfaction in active elderly. The results indicated that health, functional capacity, and environmental quality were predictors of satisfaction among the most active adults. Sella et al. [8] examined the associations between QoL (and its specific domains), objective and self-reported sleep quality, and subjective sleep-related factors (i.e., dysfunctional beliefs and attitudes about sleep and metacognitive beliefs about sleep difficulties) in healthy elderly. It was emphasized that sleep-related factors, in particular dysfunctional beliefs and attitudes about sleep, along with sleep efficiency influence the perceived QoL in healthy elderly. Thus, the elderly and quality of life have been the subject of various studies.

Amenomori et al. [10] proposed a method to continuously measure HRQoL using mobility and biometric information obtained from smartphones and smartwatches and showed that HRQoL can be estimated using a small number of questionnaires and information from smart devices. They aimed to improve HRQoL by continuously measuring HRQoL, which is strongly related to physical and mental stress states, to detect early signs of stress and to prevent it.

### 2.2. Stress Estimation

There have been many studies on stress estimation using devices in many situations. There are two groups of stress estimation studies. The first group is the studies conducted in a controlled laboratory environment [31,32]. In these studies, the researchers intentionally generated stress using some kind of stress test [33], where the researcher had complete control over the stress level, and high-stress-detection accuracy (80–97%) was usually reported.

The second group of studies analyzes stress in real life [12,34,35] and have reported relatively low accuracy [36]. Asma et al. [37] proposed the Hamilton Depression Rating Scale to assess negative psychological states. Garcia et al. [38] proposed the Oldenburg Burnout Inventory to assess stress states during work. Natasha et al. [12] proposed a method to quantitatively measure and estimate health, stress, and happiness on the next day using smart devices. Gjoreski et al. proposed the automatic detection of students’ stress using smartphones [13]. This approach aims to develop a machine learning model to detect students’ stress levels. They used data from multiple smartphones, including accelerometers, audio recorders, GPS, WiFi, call records, and light sensors.

Many studies have been conducted using biometric information, which can be acquired by wearable devices, to detect stress [18,19,39,40]. Previous approaches analyzed a combination of physiological signals, including signals collected from ECG [22], skin electrical activity [23], and electromyography [25] sensors. These approaches used traditional machine learning algorithms to analyze the physiological signals to detect stress and classify emotions. The accuracy of the results varied from 50 to 90% due to differences in environmental factors and the impact of different datasets.

Various systems are being researched to improve estimation accuracy. To analyze stress in workers, not only smartwatches, but also chest-mounted heart rate sensors (e.g., WHS-3 [41]) have been used [42]. However, systems that use multiple wearable sensors place a high burden on users and are not suitable for real-time stress detection in daily living [43]. In this study, we used as few and unobtrusive wearable devices as possible.

### 2.3. Recognition of Activities of Daily Living in the Home in a Smart Home

Georgia Institute of Technology [44], University of Colorado at Boulder [45], Microsoft Research [46], and others have initiated smart home projects on the perception of activities of daily living (ADLs) in the home. The MavHome project has conducted anomaly detection on health datasets to check for outliers and drifts in smart homes [47]. This approach is based on the regression and correlation of numerical-based health datasets. Experiments with health datasets included learning and estimating trends in human vital signs over time, e.g., whether they are increasing, decreasing, or constant. As the accuracy of such research on ADLs’ sensing within the smart home increases, it is becoming more common to apply these technologies to health support. However, in the field of health status estimation, estimations are often made with data that can be obtained from smartphones and ECG sensors. Considering the development of the health support field within smart homes, health status estimation combined with ADLs’ sensing data is an important research area. Therefore, it is necessary to conduct research combining in-home activities acquired by ADLs’ sensing and health status estimation.

### 2.4. Health Management in Smart Home

Since the elderly spend more time at home than out of the house, many systems have been developed to manage health conditions using activity data in the home. Some smart home research has attempted to correlate activities and estimate the well-being of the occupants within a living space. The goal of Intel Research’s Computer-Supported Coordinated Care (CSCC) project [48] is to identify the care network characteristics and needs of elderly people who wish to live at home. Jakkula et al. [49] aimed to identify trends in health status to create a smart home system.

However, most existing studies related to stress estimation in smart homes have been conducted in a controlled smart home, not in the subject’s home or other ordinary residences. Therefore, few studies have analyzed and estimated real-life stress in a home. In addition, in most related studies described above, the sensor data and feature values used do not consider the subject’s life in the home. Therefore, if there is no information related to daily living activities, it is impossible to understand the activities that cause stress, and it may not be possible to improve the stress state. Therefore, it is necessary to incorporate data related to activities and biometric information in homes.

## 3. Stress Estimation Method Using Biometric and Activity Indicators

### 3.1. Overview of the Proposed Method

We propose a method for estimating the stress state at the end of a day (or the beginning of the next day) from the values of the stress indicators for each of the daily living activities performed in the day using machine learning.

Figure 1 shows an overview of the proposed method. The proposed method consists of three parts. The first part is the data collection part. In this part, the biometric data and activity data are collected using smartwatches and activity recognition systems, respectively. The second part is the feature extraction part. In this part, the features are extracted from the data collected in the data collection part. The last part is the stress estimation part. The stress estimation model is constructed using machine learning. The details are described in the following sections.

### 3.2. Data Collection

The stress state can be estimated by analyzing biometric data such as heart rate variability. Heart rate variability can be measured by a wearable biometric sensor such as a smartwatch. Existing studies showed that the heart rate interval (RRI) and the standard deviation of the RRI are important indices used in stress estimation [50,51]. Therefore, we focused on the RRI variance and Lorenz plot area as stress indicators that can be calculated from heart rate variability data. Toyofuku et al. [14] proposed a simple method for estimating parasympathetic activity using Lorenz plots (LPs). A Lorenz plot is plotted on a two-dimensional plane at (RRI(t),RRI(t+1)), where RRI(t) is the RR interval at time *t*. Figure 2 shows an example of an LP. The area of the output ellipse represents the magnitude of fluctuations in the RRI and has been used and evaluated as a method to determine whether the parasympathetic nervous system has become active [27,52,53].

The equation to convert heart rate to the RRI is shown in Equation (Equation 1).
(1)$$RRI=\frac{60}{Heartrate\times 1000}$$
where RRI is defined as the heartbeat interval in ms and Heartrate is the number of heartbeats per minute. The ellipse area that covers the plotted dots is calculated as the Lorenz plot area, as shown in Figure 2.

On the y=x axis, let *m* denote the mean of the distance from the origin (0,0) and let $\delta_{x}$ denote the standard deviation from the origin. On the y=−x axis, let $\delta_{\left(-x\right)}$ denote the standard deviation from the origin (0,0). In this case, the area *S* of the ellipse with the major axis $\delta_{x}$ and the minor axis $\delta_{\left(-x\right)}$ is calculated as follows:(2)$$S=\pi \cdot \delta_{x}\cdot \delta_{\left(-x\right)}$$

We assumed an environment in which the activities of the residents can be automatically obtained by using an in-home daily living activities recognition system (such as SALON [54]). Table 1 shows an example of the activities log. The targets of the activities log are primary daily living activities such as cooking, eating meals, resting, working, cleaning, washing, going out, bathing, and sleeping. These activity logs will be recorded in the future by an in-home activity estimation system.

### 3.3. Feature Extraction

The proposed method calculates the RRI variance and Lorenz plot area for each activity from the heart rate data during each activity by the method described in Section 3.2. If the same type of activity is performed multiple times during some intervals with different timings, the biometric data measured during those intervals are integrated, and the stress indicators are calculated for the integrated data. Finally, each activity’s stress indicator value is calculated and recorded as shown in Table 2.

Residents may feel low stress when a high-stress activity is performed for a short interval and feel high stress when a low-stress activity is performed for a long interval. Thus, a combined index of time and stress intensity is necessary to express the state of stress. We create an *activity indicator* as an index of time and a *biometric indicator* as an index of stress intensity.

The *activity indicator* is defined based on the total time spent performing each activity. The duration of the activity of housework differs greatly depending on gender. Therefore, we normalized the indicators by gender. The activity indicator Ai is defined as follows:(3)$$\begin{array}{c} Ai_{s,a}=\frac{Ar_{s,a}}{\sum_{n=1}^{N}Ar_{s_{n},a}÷N} \end{array}$$
where Ar is the activity time ratio per activity in 24 h, *N* is the number of subjects, *s* is each subject, and *a* is each activity.

The *biometric indicator* is defined based on the Lorenz plot areas.In the case of Japan, the Lorenz plot of men for the cooking activity is likely to be low because some of them do not engage in any cooking activity. Therefore, we normalized it for each gender separately. Hence, we define the biometric indicator Bi as follows:(4)$$\begin{array}{c} Bi_{s,a}=\frac{Br_{s,a}}{\sum_{n=1}^{N}Br_{s_{n},a}÷N} \end{array}$$
where Br is normalized values for the Lorenz plot area per activity in 24 h, *N* is the number of subjects, *s* is each subject, and *a* is each activity.

By multiplying these two indicators, we created a *mixed indicator*, which includes both the length of time and the intensity of stress. We define the mixed indicator *MixedIndicator* as follows:(5)$$\begin{array}{c} MixedIndicator{\left(Ai,Bi\right)}_{s,a}=Ai_{s,a}\times Bi_{s,a} \end{array}$$
where Ai is the activity indicator, Bi is the biometric indicator, *s* is each subject, and *a* is each activity.

### 3.4. Stress Estimation

The extracted features were used to estimate the resident’s stress. The model for estimating stress was constructed using machine learning. The machine learning algorithm used to construct the model was Random Forest.

Random Forest is a machine learning algorithm based on ensemble learning. It uses multiple decision trees as weak classifiers and obtains the classification result by integrating the results by the weak classifiers. It is said to have higher performance with a shorter computation time for particular targets than other algorithms. Since the proposed method was designed to be processed by smart phones in the future, Random Forest was chosen because of its high accuracy and low processing load.

## 4. Evaluation Experiment

Evaluation experiments were conducted to evaluate the effectiveness of the proposed method.

### 4.1. Data Collection Experiment

The experimental data were collected from the five elderly households consisting of one single and four married couples, all in their 60s. The dataset consisted of biometric data, daily living activities data, and stress state data.

In order to collect the data, the SALON system [54] was set up in each home, which consists of a smartwatch, up to 10 motion sensors, up to 10 ambient sensors, a few door sensors, five annotation buttons, and a server. The data collection period was one month for each household.

The heart rate data as the biometric data were collected by the Fitbit Alta HR. The Fitbit Alta HR collects the heart rate once every 15 s. We converted the collected heart rate to the RRI and generated the variance of the RRI and the Lorenz plot area as the features for the stress estimation. Table 3 shows an example of the biometric indicator value for each activity based on Equation (Equation 4).

We collected data on five typical daily living activities, bathing, cooking, eating, going out, sleeping, and other. Residents recorded the start and the end of each activity by pressing the annotation buttons shown in Figure 3 installed at the locations where the activities were performed.

Table 4 shows an example of the activity indicator value based on Equation (Equation 3).

Stress state data were collected by asking questions related to physical stress after waking up and before bedtime each day. A five-point Likert scale was used in the answer. The specific question items are shown below:MQ (morning question): Did you feel physically refreshed this morning?NQ (night question): Do you experience any physical stress due to physical pain or discomfort?

### 4.2. Construction of Stress Recognition Model

#### 4.2.1. Overview of features used for stress recognition model

The feature overview is shown in Figure 4.

The data shown in Section 3.2 become the explanatory variable in the following flow. The biometric data obtained by the Fitbit were first converted to the RRI, which becomes an explanatory variable as the “average RRI”, and the “Lorenz plot area” is calculated. The Lorenz plot area is used as the “average Lorenz plot area” and “Lorenz plot area per activity”. The activity data obtained by the activity recognition system are the “time per activity” and “sleep duration”. “Sleep duration” is the same as “Time per activity of sleep”. The “mixed indicator” is calculated from the “Lorenz plot area” and “activity data”. The details are given in the following sections.

#### 4.2.2. Detail of Stress Recognition Model

The collected data were used to construct a stress estimation model. The stress estimation model was constructed by Random Forest [55]. In constructing the models, the answers on 5-point Likert scale were reorganized into three levels: good, bad, and neutral. The answers were imbalanced data. To address this issue, pre-processing such as using SMOTE-OverSampler [56] (https://imbalanced-learn.org/stable/references/generated/imblearn.over_sampling.SMOTE.html, accessed on 18 December 2022), RandomUnderSampler (https://imbalanced-learn.org/stable/references/generated/imblearn.under_sampling.RandomUnderSampler.html, accessed on 18 December 2022), and bagging was performed.

The parameters of the constructed model are shown below. RandomForestClassifier is the default setting value from scikit-learn. The upsampling parameters were sampling_strategy = 1:100, 2:100, 3:200, k_neighbors = 3, and random_state = 0. The downsampling parameters were random_state=seed and replacement = True. There were 10 models created by increasing the “seed value” by 1 between 1 and 10 with 10 times of bagging. The predicted model’s most-frequent value was used as the output of the model.

Figure 5 shows the analysis target periods (i.e., the time interval for extracting the features) to estimate the answers for morning and night questions.

In this experiment, the models of two baseline methods, one previous method, and two proposed methods were constructed. Table 5 shows the features used in each method. Baseline Method 1 used the following four basic features:Average RRI value in the last 24 h;Average RRI for 4 h after waking up;Lorenz plot area of the day;Lorenz plot area of the day for 4 h after waking up.

Baseline Method 2 adds sleep duration, which was shown to be effective in [11], to the features of Baseline Method 1. The previous method [26] adds the Lorenz plot area per activity and the Lorenz plot area per activity for 4 h after waking up to the features of the Baseline Method 2.

Proposed Method 1 adds time per activity and time per activity for 4 h after waking up to the features of the previous method. Proposed Method 2 adds the mixed indicator to the features of the Proposed Method 1.

The model was validated using the three-fold cross-validation method. The accuracy and F1-measure were used as the evaluation metrics.

The accuracy and F1-measure are defined by Equation (Equation 6) and Equation (Equation 7), respectively.
(6)$$\mathrm{Accuracy}=\frac{\mathrm{Number}\;\mathrm{of}\;\mathrm{correct}\;\mathrm{estimations}}{\mathrm{Total}\;\mathrm{number}\;\mathrm{of}\;\mathrm{estimations}}$$
(7)$$F1-\mathrm{measure}=\frac{2\mathrm{Recall}\cdot \mathrm{Precision}}{\mathrm{Recall}+\mathrm{Precision}}$$

## 5. Results

### 5.1. Result of Introducing SMOTE-OverSampler, RandomUnderSampler, and Bagging

We applied five data-processing schemes to improve the model accuracy: NotManipulated, OverSampler, UnderSampler, UnderSampler with Bagging, and Over&UnderSampler with Bagging. We used SMOTE-OverSampler and RandomUnderSampler. Bagging was performed by changing the seed value of RandomUnderSampler and made ten estimations, which were determined by the voting ensemble.

Table 6 shows the accuracy and F1-measure when applying these five schemes to our previous method [26], where the mean accuracies calculated by averaging between morning (MQ) and nighttime physical stress estimation (NQ)) were 0.65, 0.65, 0.46, 0.44, and 0.58, respectively.

In this study, the purpose was to manage the health status of the elderly. Health status management requires early detection of deterioration in the physical condition of the elderly. A minority class (the days of bad physical condition) detection was considered necessary. In Table 6, the NotManipulated and OverSampler cases showed high accuracies, but the F1-measure was low for the minority class (i.e., bad). The combination of OverSampler, UnderSampler, and Bagging gave a better accuracy and F1-measure for the “bad” class than the other methods. Although only the previous method’s result is shown in Table 6, similar results were obtained with the other evaluation methods. Therefore, SMOTE-OverSampler, RandomUnderSampler, and bagging were used for all the methods in the following evaluation experiment.

### 5.2. Effects of Different Features

We compared the five evaluation methods including the proposed methods. The result is shown in Table 7, where the estimation accuracies of 0.52, 0.55, 0.57, **0.60**, and **0.59**, respectively, were achieved for the five methods (the values were calculated by averaging the morning (MQ) and nighttime physical stress questionnaires). This suggests that adding sleep duration features (Baseline 2) somewhat improved the accuracy. Adding per-activity biometric features and the total time of daily living activity (Proposed Method 1) and the *mixed indicator* (Proposed Method 2) greatly improved the accuracy compared to Baseline 1.

Table 8 and Table 9 show more detailed estimation results. The confusion matrices for the estimation of physical stress in the morning (MQ) and at night (NQ) by all methods are given in Figure 6 and Figure 7, respectively. Proposed Method 1 shows that the number of correct answers for good increased while the number of correct answers for the others decreased or remained the same as shown in Figure 6 and Figure 7. For Proposed Method 2, the number of correct answers for good increased, as did the number of correct answers for bad and neutral.

### 5.3. Evaluation of Features by SHAP

In order to examine the effect of the feature variables on the objective variables, we used shap.Explainer of SHAP (https://shap.readthedocs.io/en/latest/index.html, accessed on 18 December 2022). Figure 8 and Figure 9 show the results of shap.Explainer. The features related to the mixed indicator were in the top 10 in terms of contribution (out of 30 features). Sleep activity is known to be important from related studies [11], and indeed, features related to sleep activity were in the top 10. Figure 8 and Figure 9 show that bathing and eating ranked high in terms of contribution, and the RRI was the highest, as stated in the previous study. The reason is that bathing helps to relieve fatigue, and eating food may be effective at reducing stress.

## 6. Discussion and Limitations

In this section, the discussion and limitations are given.

### 6.1. Discussion

The proposed method is less burdensome on users than the existing studies [13,42] because it is completed only by wearing a wearable device. Few studies combine daily living activity and biometric indicators. Furthermore, few studies use daily living activity and biometric indicators as explanatory variables to estimate health status and investigate whether they are effective as important features. In this study, heart rate data from wearable sensors and activity data from the ADL sensing system were used. The dataset used in this study consisted mostly of married couples (there was only one household with a single resident). Therefore, the time and proportion of housework was shared by two persons, and there was a variation in the duration of the activities. Hence, it was necessary to alleviate the individual differences between men and women, as well as the differences in activity duration between individuals. Therefore, we created the *mixed indicator*.

We used SMOTE for upsampling, since other upsampling methods, such as ADASYN, did not result in significant changes in accuracy, and hyperparameter adjustment using Optuna was not implemented due to its tendency to overfit due to the small number of data points. However, hyperparameter adjustment will have a good effect when a sufficient dataset is available for future large-scale experiments.

Mental stress (M2 and N2 in the questionnaire) was not targeted in this study. However, we believe that the proposed method using biological and activity indicators would also be effective to estimate mental stress. Investigating the effectiveness in estimating mental stress will be part of our future work.

### 6.2. Limitation

The ADL sensing system used in this paper assumed that the target population for sensing was relatively healthy elderly. In reality, many elderly people have cognitive and physical problems that need to be anticipated. Therefore, the main target of the proposed system is the elderly before transitioning from a healthy state to a state requiring nursing care.

In this study, only heart rate data from wearable sensors and activity data from an ADL sensing system were used to estimate the health status. The wearable sensor used was the Fitbit Alta HR, which has a coarse granularity of RRI acquisition (every 10–15 s), so rapid changes in heart rate variability cannot be obtained. Therefore, if the frequency of heart rate variability acquisition could be made more granular, there is a possibility of further improvement in accuracy. The impact of daily activities on stress may vary from person to person, because everyone’s favorite daily activities are different. Matsui et al. [24] measured biological indices of housework during daily life in the home and aimed to obtain knowledge for the establishment of a QoL estimation method by estimating stress. We also conducted a subjective evaluation of the subjects’ preferences for housework by means of a questionnaire. The results showed that the standard deviation of the RRI was lower for both subjects in all housework daily living activities (room cleaning, cooking, laundry, and dish washing) than in the other daily living activities. In addition, ID01 and ID02 indicated that they did not prefer cleaning and washing dishes, respectively, as their preferences for housework in the preliminary questionnaire. The standard deviation of the RRI of each subject was relatively low for each of the daily living activities, suggesting that there is a relationship between housework preferences and biometric indices. According to the results of the pre-survey, the LF/HF values were higher for the cooking and dish washing daily living activities in ID02 when the degree of preference for these daily living activities was high. On the other hand, LF/HF did not always coincide with the degree of preference or skill in some daily living activities, such as washing clothes and cleaning in ID01. According to the results of the pre-sleep questionnaire, each subject felt most frequently stressed by the cooking daily living activity, but the RRI tended to be relatively low and the LF/HF tended to be relatively high, reflecting subjective evaluation. Although the subjective evaluation and the stress estimated by the bioindicators did not necessarily coincide, the tendency of increased stress due to various household chores and the tendency of relaxation due to rest and entertainment were observed, suggesting that there are several relationships between daily life daily living activities and the bioindicators. Thus, the impact of daily routines on stress may vary from person to person, as each person’s favorite chore in daily life is different. However, the dataset we used in this study was collected “in the wild” from healthy elderly who were able to lead a normal daily life. Because this dataset is “in the wild”, the number of unhealthy individuals was small, and we were not able to obtain data on the domestic daily living activities among individuals. Therefore, it was not possible to specifically visualize differences in favorite daily activities among individuals. However, it is possible to do so with a dataset that has acquired a sufficient amount of data on individuals.

Some studies have been conducted using biometric information that can be acquired by wearable devices to detect stress [19,22,23,25,39,40]. The accuracy of the results varied from 50 to 90% [18,57,58] due to differences in environmental factors and the impact of different datasets. These methods sometimes result in a high-burden system for the elderly because they do not take into account the characteristics of the elderly, who are not accustomed to digital devices and tend to be strict about privacy. In this system, we created a low-burden stress estimation method because it uses data on in-home activities that can be automatically acquired by a smart house and biometric data that can be automatically collected by wearing a smartwatch. Another difficulty with our dataset is that it is “in the wild” and not rich in data. The number of housework activities was six, and the characteristics of the device were non-invasive and low-burden, rather than invasive and high-burden, to meet the needs of the elderly. The accuracy of this study was not so high, but this will be resolved with advanced devices and an increase in the types of housework activities from activity recognition systems.

### 6.3. Future Work

As future work of this study, we will create an application that has functions to estimate the activities of residents and their health status. Figure 10 shows a mockup screen of the web application. The application obtains the information on the activity of its user by ADLs’ sensing and annotation to estimate his/her health status. Then, several sequences of expected activities that will happen in the future will be listed. The sequences are classified into two classes: improving and worsening physical conditions. The application then encourages the user to change future activities that will improve their physical condition (thumbs-up icon in Figure 10).

## 7. Conclusions

In this paper, to improve the estimation accuracy of stress estimation, we proposed a method to estimate the stress level by linking daily living activities data and biometric data. Through the experiment using the dataset collected from elderly households, it suggests that adding per-activity biometric features and the total time of daily living activity (Proposed Method 1) and the *mixed indicator* (Proposed Method 2) greatly improved the accuracy compared to the baseline methods. Future work on this research will include: creating and evaluating an application with the functions to estimate the activities and health status of residents and recommend future activities for a better health condition.

## Figures and Tables

**Figure 1 sensors-23-00535-f001:**
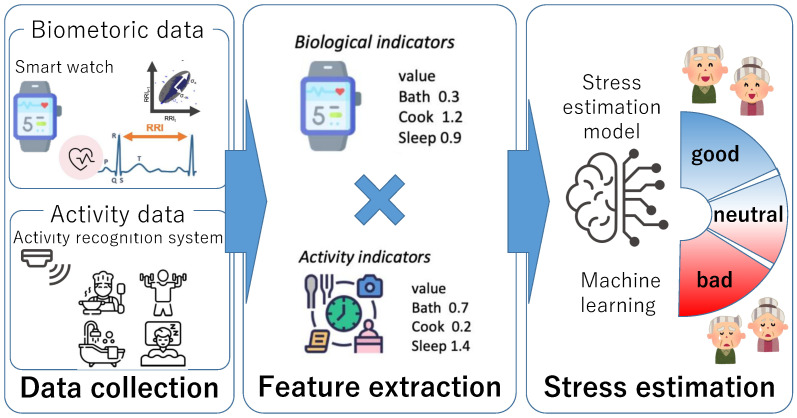
Overview of the proposed method.

**Figure 2 sensors-23-00535-f002:**
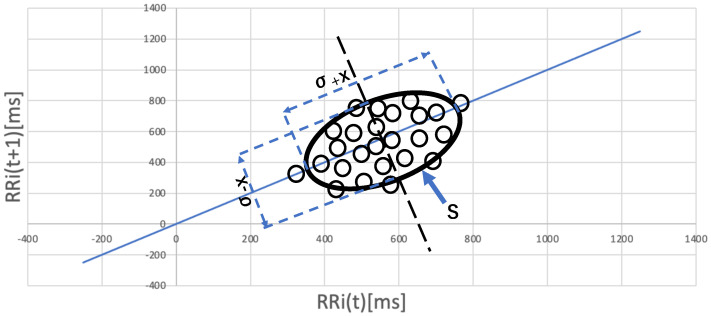
How to calculate the Lorenz plot area.

**Figure 3 sensors-23-00535-f003:**
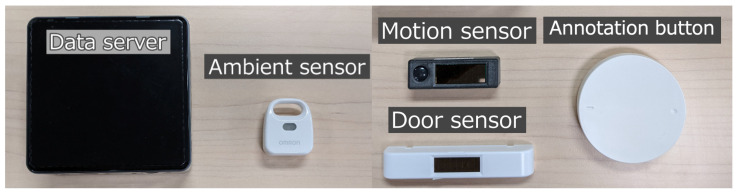
Components of the SALON system.

**Figure 4 sensors-23-00535-f004:**
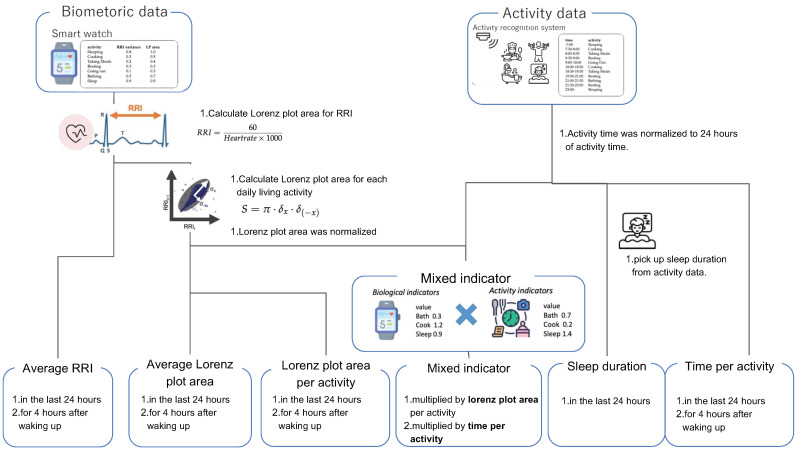
Overview of stress recognition model.

**Figure 5 sensors-23-00535-f005:**
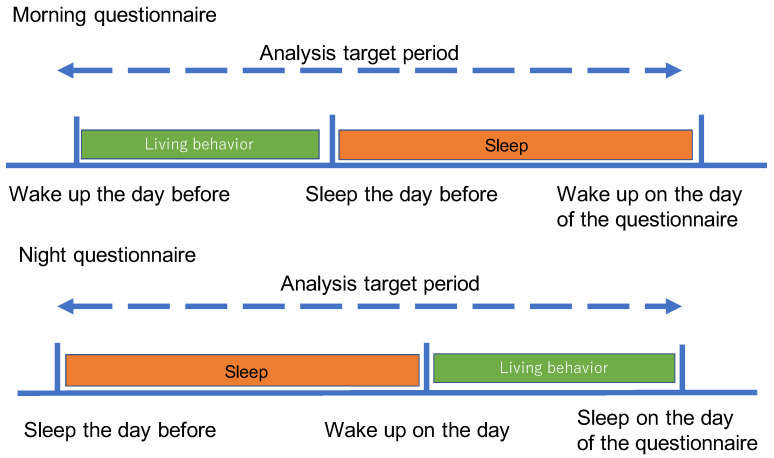
Aggregation period for morning and night questionnaires.

**Figure 6 sensors-23-00535-f006:**
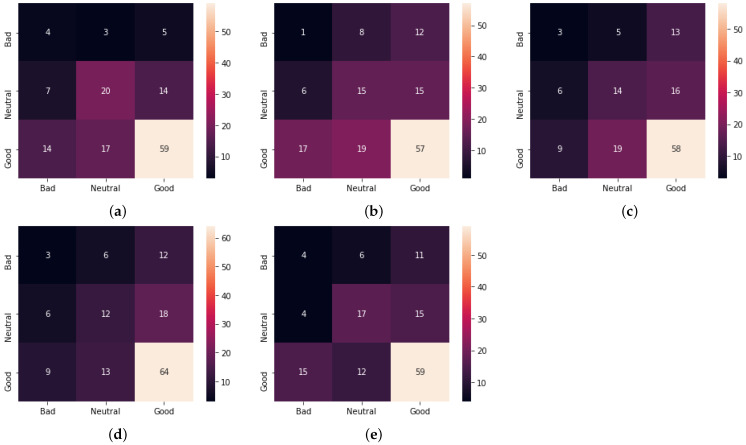
Confusion matrix for the five methods for the physical stress estimation in the morning (MQ). (**a**) Baseline Method 1; (**b**) Baseline Method 2; (**c**) Previous method [26]; (**d**) Proposed Method 1; (**e**) Proposed Method 2.

**Figure 7 sensors-23-00535-f007:**
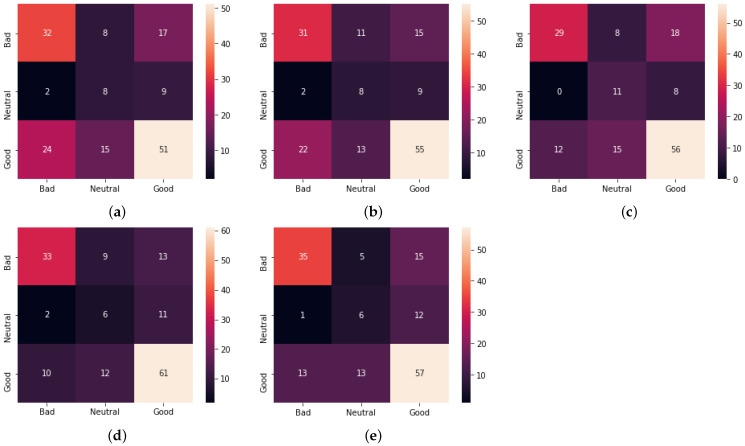
Confusion matrix for the five methods for the physical stress estimation at night (NQ). (**a**) Baseline Method 1; (**b**) Baseline Method 2; (**c**) Previous method [26]; (**d**) Proposed Method 1; (**e**) Proposed Method 2.

**Figure 8 sensors-23-00535-f008:**
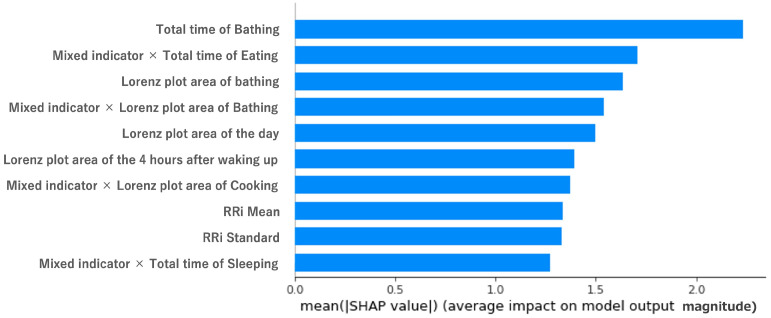
Impact of features on model output by SHAP values for MQ. Vertical axis: each feature (in decreasing order of contribution, top 10). Horizontal axis: value of the contribution variable (SHAP value).

**Figure 9 sensors-23-00535-f009:**
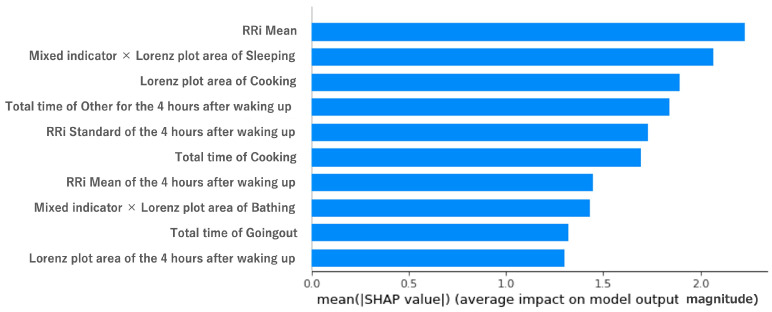
Impact of features on model output by SHAP values for NQ. Vertical axis: each feature (in decreasing order of contribution, top 10). Horizontal axis: value of the contribution variable (SHAP value).

**Figure 10 sensors-23-00535-f010:**
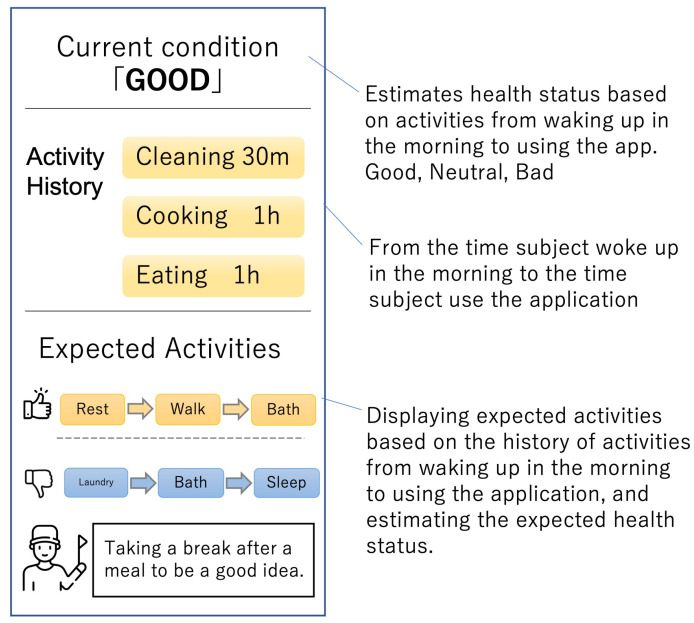
Mockup for future work.

**Table 1 sensors-23-00535-t001:** Activities log.

Time	Activity
–7:00	Sleeping
7:30–8:00	Cooking
8:00–8:30	Eating Meals
8:30–9:00	Resting
9:00–18:00	Going Out
18:00–18:30	Cooking
18:30–19:30	Eating Meals
19:30–21:00	Resting
21:00–21:30	Bathing
21:30–23:00	Resting
23:00–	Sleeping

**Table 2 sensors-23-00535-t002:** Per-activity stress indicators.

Activity	RRI Variance	LP Area
Sleeping	0.8	1.0
Cooking	0.3	0.5
Eating Meals	0.2	0.4
Resting	0.3	0.3
Going out	0.1	0.3
Bathing	0.5	0.7
Sleep	0.9	0.9

**Table 3 sensors-23-00535-t003:** Biometric indicator for Subject ID1/woman.

Activity	Value
Bathing	1.13
Cooking	1.21
Eating	0.67
Going out	0.80
Sleeping	0.95
Other	1.10

**Table 4 sensors-23-00535-t004:** Activity indicator for Subject ID1/woman.

Activity	Value
Bathing	0.40
Cooking	1.84
Eating	0.66
Going out	0.71
Sleeping	1.11
Other	1.14

**Table 5 sensors-23-00535-t005:** Specific features for each method.

	Baseline 1	Baseline 2	Previous Method	Proposed Method 1	Proposed Method 2
Basic features	X	X	X	X	X
Sleep duration		X	X	X	X
Lorenz plot area per activity			X	X	X
Lorenz plot area per activity for 4 h after waking up			X	X	X
Time per activity				X	X
Time per activity for 4 h after waking up				X	X
Mixed indicator multiplied by Lorenz plot area per activity					X
Mixed indicator multiplied by time per activity					X

**Table 6 sensors-23-00535-t006:** Result of introducing SMOTE-OverSampler, RandomUnderSampler, and bagging to the previous method [26]. Estimation values are averaged over MQ and NQ.

	Not Manipulated	OverSampler	UnderSampler	UnderSampler and Bagging	Over, UnderSampler, and Bagging
Accuracy	0.65	0.65	0.46	0.44	0.58
F1-measure					
Bad	0.27	0.37	0.41	0.43	**0.47**
Neutral	0.24	0.37	**0.38**	0.35	**0.38**
Good	**0.76**	0.75	0.54	0.49	0.66

**Table 7 sensors-23-00535-t007:** Estimation accuracy of five evaluation methods. Estimation values are averaged over MQ and NQ.

	Baseline 1	Baseline 2	Previous Method	Proposed Method 1	Proposed Method 2
MQ: Morning physical stress	0.49	0.49	0.54	0.55	**0.56**
NQ: Nighttime physical stress	0.55	0.57	0.61	**0.64**	0.62
Mean	0.52	0.55	0.57	**0.60**	0.59

**Table 8 sensors-23-00535-t008:** Estimation accuracy of five methods for MQ.

	Baseline 1	Baseline 2	Previous Method	Proposed Method 1	Proposed Method 2	Support
Accuracy	0.49	0.53	0.52	0.55	**0.56**	143
ine Recall						
Bad	0.35	0.05	0.15	0.14	0.23	21
Neutral	0.38	**0.42**	0.39	0.33	**0.42**	36
Good	0.66	0.61	0.67	**0.74**	0.70	83
ine F1-measure						
Bad	0.36	0.04	0.16	0.23	**0.45**	21
Neutral	0.34	0.38	0.38	0.36	**0.44**	36
Good	0.66	0.64	0.67	**0.71**	0.69	83

**Table 9 sensors-23-00535-t009:** Estimation accuracy of five methods for NQ.

	Baseline 1	Baseline 2	Previous Method	Proposed Method 1	Proposed Method 2	Support
Accuracy	0.55	0.57	0.61	**0.64**	0.62	157
ine Recall						
Bad	0.56	0.54	0.53	0.60	**0.64**	55
Neutral	0.42	0.42	**0.58**	0.32	0.32	19
Good	0.57	0.61	0.67	**0.73**	0.69	83
ine F1-measure						
Bad	0.56	0.60	0.38	0.66	**0.67**	55
Neutral	0.32	0.31	**0.42**	0.26	0.28	19
Good	0.61	0.65	0.68	**0.73**	0.68	83

## Data Availability

Not applicable.

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
