# Peer review of "Stress Estimation Using Biometric and Activity Indicators to Improve QoL of the Elderly†"

_sensors, 2023, doi:10.3390/s23010535_

Round 1

Reviewer 1 Report

Please refer the attached review report.

Author Response

Dear Reviewer,

Thank you for inviting us to submit a revised draft of our manuscript entitled, "Stress estimation Using Biometric and Activity Indicators to Improve QoL of the Elderly".

Sincerely,

Kanta Matsumoto

Graduate School of Information Science, Nara Institute of Science and Technology

8916-5, Takayama, Ikoma, Nara, 630-0192, JAPAN

[email protected]

+81 90-9947-8447

Reviewer 2 Report

In this paper, the authors propose a method to improve the stress estimation accuracy by linking daily living activity data and biometric data. The study is interesting and has practical significance. However, several concerns are as follows.

1.     The impact of daily activities on stress may vary from person to person, because everyone's favorite daily activities are different. This may lead to no strong correlation between daily activities and stress.

2.     The estimation accuracy of more than 5% is actually not high, compared with the inaccuracy contained in the data obtained and processed.

3.     The modeling algorithms used in the study should be explained clearly.

Author Response

(The authors gave the same response as above.)

Reviewer 3 Report

1.      Very poor English was used for the entire manuscript. Author must do complete proof read.

2.      Any manuscript, Abstract play a major role to define the proposed model. Here abstract just like information only.

3.       Lack of survey information and no recent research related to Biometric and Activity Indicators to Improve QoL.

4.      Any manuscript, the flow process or entire process would be explained as system architecture. Here I could not find any system model to understand the proposed works.

5.      How they were implemented this model ? What is the input parameters ? How the output parameters could calculated? What is the performance validation? all the above information were not described properly.

6.      Author need to work hard and improve the manuscript in better way and I hope my comments will not discourage the author works. All the best

Author Response

(The authors gave the same response as above.)

Round 2

Reviewer 3 Report

No more further comments.